# Gait Phase Detection Based on Muscle Deformation with Static Standing-Based Calibration

**DOI:** 10.3390/s21041081

**Published:** 2021-02-04

**Authors:** Tamon Miyake, Shintaro Yamamoto, Satoshi Hosono, Satoshi Funabashi, Zhengxue Cheng, Cheng Zhang, Emi Tamaki, Shigeki Sugano

**Affiliations:** 1Faculty of Science and Engineering, Waseda University, Tokyo 169-8555, Japan; sugano@waseda.jp; 2Graduate School of Advanced Science and Engineering, Waseda University, Tokyo 169-8555, Japan; s.yamamoto@fuji.waseda.jp; 3H2L Inc., Tokyo 106-0032, Japan; satoshi.hosono@h2l.jp; 4Graduate School of Creative Science and Engineering, Waseda University, Tokyo 169-8555, Japan; s_funabashi@sugano.mech.waseda.ac.jp (S.F.); hoimei@acm.org (E.T.); 5Graduate School of Fundamental Science and Engineering, Waseda University, Tokyo 169-8555, Japan; zxcheng@asagi.waseda.jp; 6Department of Mechanical Systems Engineering, Ibaraki University, Ibaraki 316-0033, Japan; cheng.zhang.abbott@vc.ibaraki.ac.jp; 7Global Information and Telecommunication Institute, Waseda University, Tokyo 169-8050, Japan

**Keywords:** gait phase detection, muscle deformation, static standing-based calibration

## Abstract

Gait phase detection, which detects foot-contact and foot-off states during walking, is important for various applications, such as synchronous robotic assistance and health monitoring. Gait phase detection systems have been proposed with various wearable devices, sensing inertial, electromyography, or force myography information. In this paper, we present a novel gait phase detection system with static standing-based calibration using muscle deformation information. The gait phase detection algorithm can be calibrated within a short time using muscle deformation data by standing in several postures; it is not necessary to collect data while walking for calibration. A logistic regression algorithm is used as the machine learning algorithm, and the probability output is adjusted based on the angular velocity of the sensor. An experiment is performed with 10 subjects, and the detection accuracy of foot-contact and foot-off states is evaluated using video data for each subject. The median accuracy is approximately 90% during walking based on calibration for 60 s, which shows the feasibility of the static standing-based calibration method using muscle deformation information for foot-contact and foot-off state detection.

## 1. Introduction

### 1.1. Background

Chronic diseases caused by ageing are associated with a decline in walking ability [1,2]. Early-discovery of the decline in gait speed or the increase in the variability of the gait cycle duration is important for identifying prodromal dementia as a chronic disease [3,4]. In addition, gait-assistance devices have been developed for mitigating the impairments due to the decline in walking ability [5,6]. Gait phase detection is essential for monitoring human health conditions and for synchronous gait assistance with devices.

Gait can be classified into two phases: stance and swing. As shown in Figure 1, foot-contact and foot-off states during the gait cycle are defined as the stance phase and swing phase, respectively [7]. In the stance phase, the foot-contact posture changes according to heel contact, foot flat, and heel off. The swing phase starts with toe-off and ends with heel contact. A camera system is used as the gold standard method to detect foot-contact or foot-off states [8]. However, the system cannot be used outdoors for gait phase detection because of the lack of portability. This is crucial for widespread daily use. Therefore, wearable sensors for the classification of gait phases have been proposed for monitoring human gait motions in daily life.

### 1.2. Previous Research

Gait phase detection methods have been investigated with sensing techniques based on foot contact force or lower limb kinematics for decades. One of the representative sensors for gait phase detection is the force-sensing resistor (FSR), which is a thin film force sensor that can be used as a foot switch. Classical methods set a threshold using FSRs [9,10,11], the F-scan^®^ system [12], or insoles with FSRs [13]. To ensure adaptability to individuals and compensate for gait condition differences, various approaches have been developed to improve the calculation accuracy. For instance, an automatic threshold-tuning algorithm with FSRs was proposed [14,15]. Using such insole sensors, a proportion-based fuzzy algorithm or hidden Markov model (HMM) was proposed [16,17]. However, film sensors attached to the sole of the foot still experience severe durability issues because the repeated foot contact strongly influences the sensing reliability [13,18].

In addition to FSRs, different types of sensors have been used to provide kinematic information for gait phase detection while avoiding foot-contact impact to the sensors. For example, inertial measurement units (IMUs) have been widely used with HMM [19,20,21], support vector machine (SVM) [22], Gaussian mixture model [23], convolutional neural networks [24], and long short-term memory-deep learning [25]. Electromyography (EMG)-based methods have been proposed for gait phase detection using SVM [26], artificial neural networks [27], and linear discriminant analysis (LDA) [28]. Force myography information, which can measure muscle activity with less sensitivity to sweat than EMG, was also used for gait phase detection with LDA [29]. Joint angular sensors of lower-limb exoskeletons were used with multi-layer perceptron [30]. However, the above techniques are based on supervised machine learning algorithms, which require manual calibration to collect a significant amount of precise training data and corresponding label data, such as the video frames from ground truth, during the training phase. The calibration work with walking data burdens the users, and thus significantly limits their usability.

Several approaches are available for gait phase detection without obtaining training data. An auto-calibration method using a planar structure in the angular space of leg joints was proposed [31] to detect heel contact and toe-off. However, this method requires expert knowledge to attach three expensive joint angle sensors to the subjects, thereby increasing the cost, time of attachment, and complexity of use. An IMU-based system could apply a rule-based detection algorithm based on thresholds or searching for minimum and maximum values, which do not require training data [32,33]. For instance, a zero-crossing detection approach finding positive and negative peaks of angular velocity detected gait events within an error of 103 ms [32]. An IMU-based system with a rule-based algorithm using dynamic image features was used for identification of gait biometrics [34]. Rule-based methods are limited to inertial information among wearable devices; other sensing information might require machine learning because the sensor readings vary due to differences in the attachment conditions of the sensor. Using different sensing information for gait phase detection without manual labeling the gait data would hep widen the application scenarios or contribute to future fusion of sensing information for increased detection accuracy.

### 1.3. Objective

In this study, we focused on muscle deformation during walking to construct a novel method of gait phase detection. The objective of this study was to establish a gait phase detection system for foot-contact/foot-off state detection based on muscle deformation using a machine learning algorithm with fast and easy semi-automatic calibration and does not require manual labeling using gait data. Our hypothesis was that static standing-based calibration performed for a short time, just before walking, enables the user to use the calibrated sensor system for gait phase detection while avoiding labeling work. Our previous research demonstrated the potential of muscle deformation information sensed by only one sensor for gait phase detection with calibration using static standing posture data [35]. The hyperparameters were adjusted and different for each subject; however, they cannot be adjusted with standing data. Although the decision boundary detecting the foot-contact/foot-off states might not be perfectly consistent between standing and walking data without adjusting hyperparameters for each individual, we assumed that a probability can be adjusted from simple information about the direction of leg movement (anterior or not). In this paper, we present a probability adjustment method based on leg motion information as an alternative method to hyperparameter adjustment.

Calf muscle deformation data were monitored because calf muscles, which are related to ankle movement, have a strong relationship with foot-contact information [7]. We investigated combinations of postures, in which measured calf muscle deformation enabled the system to classify stance and swing phase during walking, and we evaluated the accuracy of the proposed system. A logistic regression (LR) algorithm that can calculate class probability was used as a machine learning algorithm and evaluated using an effective combination of training standing data.

The contributions of this work are as follows:Establishing an easy and fast semi-automatic calibration method for gait phase detection using discrete static standing data with muscle deformation information;Enabling the use of a gait phase detection system with only one sensor.Conducting experiments with the novel gait phase detection system on humans.

## 2. Proposed System

In this section, we present a detection system for foot-contact/foot-off based on LR, the probability of which is adjusted using the leg kinematic information with static standing-based calibration using muscle deformation information. Figure 2 depicts an overview of the system. The details of the sensor and classification algorithm are explained in the next subsections.

### 2.1. Sensor

FirstVR (model number: FVR-SET01, year of manufacture: 2018) [36] was used as the muscle deformation sensor, as shown in Figure 3. The muscle deformation sensor consists of a photoreflector with a set of LEDs that emit near-infrared light installed near the body surface and a photodiode that senses the intensity of infrared rays. The photoreflector is installed on the body surface to measure the skin deformation (muscle deformation) caused by the muscle bulge. The muscle deformation sensor should be installed at a distance of approximately 5 mm from the body surface, and the deformation of the skin (muscle deformation) caused by the bulge of the muscle can be measured with a change from 0.5 to 5 mm. In addition to the infrared LED part and light-receiving part, the muscle deformation sensor is equipped with a suction-cup-type case that surrounds the light-emitting part and the light-receiving part with a circumference of 8–10 mm (diameter). In this experiment, a cushioning material was attached around the muscle deformation sensor to facilitate the sensor’s response to muscle deformation. FirstVR is a wearable device with 14 muscle-deformation sensors installed in series on a belt. By wrapping the belt around an arm or a leg, the ridge of the muscle (muscle displacement) around the arm or leg can be measured. FirstVR is also equipped with magnetometric and gyroscope sensors. These sensing values were transmitted to a computer using Bluetooth low energy.

### 2.2. Classification Method

Our system is based on a machine learning algorithm with static standing-based calibration for the binary classification of gait phases. Calf muscle deformation data were obtained while standing with several postures of gait motion, which trained the machine learning algorithm. Muscle deformation is caused by muscle contraction when a joint is moved or weight is applied. Because the calf muscle acts in conjunction with foot motion, training standing data corresponding to foot position were selected. Here, 7 postures were considered as training data options in terms of foot position (backside or front-side) and foot-contact/foot-off states, as shown in Figure 4. These postures were: foot-flat in backward position, heel-off, foot-off in backward position, foot-flat in forward position, heel-contact, foot-off in forward position, and single support. We investigated which posture was effective for the training of the algorithm classifying swing and stance phases.

LR is used for the binary classification algorithm of the gait cycle with static standing-based calibration in this study. Because the algorithm should be trained quickly to ensure usability, and the amount of input data is limited, a deep learning method that is adaptable to the complicated structures of huge inputs and requires time to be trained is not appropriate. SVM can be considered as a high-ability binary classification algorithm. However, because the classification ability of SVM relies on hyperparameters related to the penalty weight of the error and the scale of the feature space, which can differ between users, classifying gait muscle deformation data would be difficult using SVM in a static standing-based calibration method, where cross-validation (hyperparameter tuning) cannot be performed for each individual. However, LR does not adjust the scale and penalty weight as hyperparameters. Furthermore, LR can output the probability of respective classes. Even though the muscle deformation state during standing might be similar to the muscle deformation state in some phases during walking, it is challenging for the decision boundary to be perfectly consistent between standing and walking. Therefore, we determined that probabilistic consideration was required. Deriving probability from muscle deformation information enables the system to reflect another type of sensor information for the adjustment of probability. In this study, we used the simple information of leg movement obtained with FirstVR for adjustment of probability.

LR is an algorithm that classifies input vectors into two classes using probability. The probability is calculated based on the inner product value of the weight vector **w** and the input vector **x**, which is derived for solving a linear regression problem. The LR algorithm calculates the regression value by assigning this inner product to the sigmoid function, which is obtained from 0 to 1 as probability *P* as:(1)$$P=\sigma \left(w^{T}x\right)=\frac{1}{1+exp(-w^{T}x)}.$$

The decision boundary is a region of values where *P* is 0.5; that is, $w^{T}x$ is zero. In this study, the class was defined as the swing phase if *P* was more than 0.5 and as the stance phase if *P* was less than 0.5.

Because the decision boundary might slightly differ between standing and walking data, the gyroscope and magnetometer values from FirstVR were used for adjusting the decision boundary. The probability of the swing phase is higher when the leg moves in the anterior direction of the body. Therefore, the size of the internal product $w^{T}x$ was larger or smaller, corresponding to the angular velocity of the sensor (shank) in the forward direction compared with a threshold. Considering the drift of the gyroscope, the threshold value was set based on the gyroscope values during standing. First, the Euler angle of the sensor was obtained by the quaternion of the magnetometer, and was derived based on [37] as:(2)$$\left[\begin{array}{c} \phi \\ \theta \\ \;\psi \end{array}\right]=\left[\begin{array}{c} arctan\frac{2(q_{0}q_{1}+q_{2}q_{3})}{q_{0}^{2}-q_{1}^{2}-q_{2}^{2}+q_{3}^{2}}\\ arcsin(2\left(q_{0}q_{2}-q_{1}q_{3}\right))\\ arctan\frac{2(q_{0}q_{3}+q_{1}q_{2})}{q_{0}^{2}+q_{1}^{2}-q_{2}^{2}-q_{3}^{2}} \end{array}\right],$$
where $q_{0}$, $q_{1}$, $q_{2}$, and $q_{3}$ are the quaternions of FirstVR.

Next, the angular velocity in the spatial coordination system was calculated from the angular velocity ω along the axis in the relative coordinate system of the sensor (reading values of the gyroscope) as:(3)$$\left[\begin{array}{c} \dot{\phi }\\ \dot{\theta }\\ \;\dot{\psi } \end{array}\right]=\left[\begin{array}{ccc} 1 & \sin\phi \tan\theta & \cos\phi \tan\theta \\ 0 & \cos\phi & -\sin\phi \\ 0 & \sin\phi \sec\theta & -\cos\phi \sec\theta \end{array}\right]\left[\begin{array}{c} \omega_{x}\\ \omega_{y}\\ \omega_{z} \end{array}\right],$$
where ϕ, θ, and ψ are the roll, pitch, and yaw in the spatial coordinate system, respectively. Using the derived angular velocity in the spatial coordinate system, the adjusted *P* was derived as:(4)$$P=\left\{\begin{array}{cc} \frac{1}{1+exp(-\left(w^{T}\phi +a\right))} & \left(gy<th1\right)\\ \frac{1}{1+exp(-\left(w^{T}\phi -a\right))} & \left(gy>th2\right)\\ \frac{1}{1+exp(-w^{T}\phi )} & \left(otherwise\right) \end{array}\right.,$$
where *a* is the adjustment value, gy is the angular velocity related to the anterior direction of the body, and th1 and th2 (which is higher than th1) are the threshold values of the angular velocity. We assumed that the foot moved in the forward direction if gy is less than th1, while it does not move in the forward direction if gy is more than th2. The set values of *a*, th1, and th2 in this paper are explained in the experimental section. This algorithm is referred to as LR-a in this paper.

## 3. Experiment

We conducted an experiment to evaluate the detection ability of the proposed gait phase detection system based on static standing-based calibration. The gait data were classified into two labels: the stance (foot-contact state) and swing (foot-off state) phases. We recruited 10 subjects, compared the estimated labels and ground-truth label obtained from video, and evaluated the effect of the standing data combination, algorithm, and data amount on the detection accuracy.

### 3.1. Data Acquisition

Ten healthy adults (six men and four women; age: 26.2±2.36 years; height: 1.67±0.12 m) were recruited; none had any neurological injuries or gait disorders. Before obtaining their consent, we explained the detailed objective of this study, and participants were allowed to withdraw from the experiment whenever they desired. The experiment was conducted with the approval of the Institutional Review Board of Waseda University (No. 2020-262).

We first obtained muscle deformation data using FirstVR attached to the center of the right lower leg in the long axis direction, to be used as training data. The subjects were asked to assume seven postures, as shown in Figure 4, in terms of foot position (backside or front-side) and foot-contact state. We guided the subjects to stand in these postures by demonstrating them ourselves. Thereafter, we measured walking data on a treadmill for system evaluation. We recorded videos obtained from the right side of the body as well as muscle deformation data with FirstVR. In addition, the quaternion and angular velocity of FirstVR were measured by its installed magnetometric and gyroscope sensors. The sampling frequencies of the camera and FirstVR systems were 120 and 50 Hz, respectively. The video and FirstVR system were synchronized by videotaping the FirstVR connection interface at the start of the measurement. The subjects walked at their preferred speed (2.93±0.26 km/h) on the treadmill for approximately 1 min, and 30 gait cycle data were extracted. In total, 19,651 frames were used for the evaluation.

### 3.2. Considering the Combination of Standing Data

Because all combinations of standing data for training the algorithm were enormous, we specified the combination conditions. To consider the combination of standing data as experimental conditions, the muscle deformation vectors of standing and walking were compared. The Euclidean distance between the mean muscle deformation vectors of each posture and the muscle deformation vectors of any phase during walking was derived, and the number of gait labels (i.e., stance or swing phase) where the Euclidean distance was the smallest for each posture was counted in each gait cycle. The percentages of the counted closest labels were derived by dividing the counted numbers of stance and swing phases by the total number of gait cycles in each posture. The strength of the relationship between the phase and posture data increased with the percentage of labels in the posture data.

Figure 5 shows the percentages of labels whose muscle deformation vectors were closest to those of each posture in each gait cycle. The muscle deformation data of back foot-flat were the most similar to the data of the swing phase. Although front foot-off is a swing-phase posture, the percentage of stance phase was higher than for swing phase. The percentage of the stance phase was in the descending order of: front heel-on, single support, back foot-flat, front foot-flat, and back heel-off among the stance phase postures. The training standing data were included in order of Euclidean distance from the walking data of the stance or swing phases as training data combinations, as shown in Table 1.

### 3.3. Evaluation of Gait Phase Detection

The system was evaluated with sample-based accuracy as:(5)$$Sample\;based\;accuracy=\frac{N_{true}}{N},$$
where $N_{true}$ and *N* denote the number of correctly classified samples and the total number of evaluation samples, respectively. We manually annotated the ground-truth labels (i.e., stance or swing phase) by viewing the video. Frames in which the heel of the right foot contacted the treadmill and the toe of the right foot left the treadmill were extracted by visual observation. The frames from the toe-off to the heel-contact and from the heel-contact to the toe-off were labeled as the swing phase and stance phase, respectively. Among the 14 channels of FirstVR, 11 were used as model inputs because the other 3 channels did not work. The model was trained with the standing data (Figure 4) for each subject. The accuracy was derived for each subject, and a box-plot was constructed. The data distribution of the accuracies was non-parametric.

LR, SVM with tuned hyperparameters, and LR-a (adjusted LR using angular velocity data) were used as the detection algorithm. These algorithms were implemented in Python using a scikit-learn library. The LR solver was newton-cg and the other hyperparameters were the default. The best *C* value of SVM, which is related to penalty weight, was selected among 1, 10, 100, and 1000; and the best γ value of SVM, which is related to scale of the data space, was selected among 0.0001, 0.001, 0.01, and 0.1 for each person. Other hyperparameters of the SVM were left as default. The adjustment parameter *a* of Equation (Equation 4) was considered to range from 0 to 10, and 8 was used because it resulted in the highest accuracy. Figure 6 shows the angular velocity related to the anterior direction of the body during walking. We set thresholds Th2 and Th1 as 0.7 and 0.6, respectively, based on the angular velocity value during static standing. In the case where the attachment orientation of the sensor was upside down for some subjects, the angular velocity value was multiplied by −1. Because a large error of transformation (especially tangent calculation) causes a fault in the probability adjustment, the angular velocity values were unreliable when tan(θ) was significantly high. In this study, the adjustment was not performed if the maximum tan(θ) was more than 50.

First, the training data combination required for gait phase detection was investigated. Table 1 shows the experimental conditions of the standing data used for calibration. These 10 training datasets were selected based on the Euclidean distance from the data of the swing or stance phase, as explained above. Because SVM has a strong binary classification ability, the effect of 10 training datasets on detection accuracy with SVM was investigated. The Kruskal–Wallis test was performed to evaluate the significance of the effect of the training data combination on accuracy. Statistical analyses were performed using the MATLAB toolbox. The significance level was set to 0.05.

Next, the effect of algorithms of LR, SVM, and LR-a on the detection accuracy was evaluated. To consider the duration that people should stand for calibration, the detection results for data numbers of 500, 1000, 1500, and 2000 for each posture were compared. The Friedman test in the MATLAB toolbox, which is a non-parametric version of analysis of variance, was performed to evaluate the effect of algorithms and data amount. The significance level was set to 0.05.

The detection results for each phase were evaluated using experimental conditions for accuracy. True positive rates for the swing and stance phases were derived. The true positive rate is the proportion of frames that were correctly identified as positive, which is calculated as:(6)$$True\;positive\;rate=\frac{N_{phasetrue}}{N_{phase}},$$
where $N_{phasetrue}$ and $N_{phase}$ denote the number of correctly classified and total data for a certain phase, respectively.

## 4. Results and Discussion

Figure 7 shows the accuracy produced by SVM for each training data combination explained in Table 1. The Kruskal-Wallis test showed that the differences in the training data combinations produced significant differences in detection accuracy. Notably, samples of conditions 6 and 9 were significantly different.

All conditions included the postures of the back and front heel contacts because the muscle deformation vectors of these postures were close to those of the stance or swing phase, respectively. We assumed that these two postures were key factors for identifying foot-contact and foot-off states during walking. The postures of the back heel-off and front foot-off, whose muscle deformation vectors were farthest from those of the swing or stance phase, could be factors that reduce the detection accuracy. However, the conditions including the posture of back foot-off, such as condition 9, tended to result in higher accuracy, except for condition 6. Because of the arbitrary nature of human posture, differences may exist in the posture between subjects. Even if the position of the foot of a certain posture is slightly different, the resulting change in muscle deformation status may be sufficiently large to affect the accuracy of gait phase detection. For several subjects, the accuracy tended to be lower when only the muscle deformation data of the back foot-off posture was used as the training data for the swing phase. We assumed that foot position during standing in the back foot-off of these subjects was different from that of other subjects. This might reduce the number of postures required for calibration if the arbitrary posture can be defined in more detail. However, three to five postures were required as training data to compensate for the arbitrary values in this work. Because the median accuracy in condition 9 was the highest, we compared the effect of the differences in the algorithm and training data number using condition 9.

Figure 8 shows the detection results when using LR, SVM, and LR-a with 500, 1000, 1500, and 2000 training data for each posture. The Friedman test showed that there was a significant difference between algorithms, but there was no difference between amount of data for each posture. LR-a produced the highest accuracy. Figure 9 shows an example time-series result of gait phase detection of a male subject using LR-a. Table 2 shows the true positive rates of the stance and swing phases detected by LR-a. The false positive rate of a certain phase, which is the proportion of positives that are wrongly counted, is equal to the value derived by subtracting the true positive rate of the other phase from 100%. Therefore, the median false positive rates of the stance and swing phases were approximately 18% and 6%, respectively. Because there were approximately 1.5 times more stance phase samples than swing phase samples, the mismatching count was almost the same between phases. Therefore, we assumed that the detection ability of the proposed method is not biased toward either the swing phase or the stance phase.

This study is the first to examine a gait phase detection method with static standing-based calibration using muscle deformation information. The static standing-based calibration enabled the system to be calibrated without walking. As shown in Figure 8, the median accuracy is 90% in the best condition. Although one of the previous rule-based methods with IMU [33] did not evaluate sample-based accuracy, the detection accuracy of such methods might be same as this work considering absolute mean error and ratio of strides where gait events were detected to all strides in the studies [32,33]. The literature using machine-learning-based methods reports higher accuracy than rule-based methods and our method. The accuracy of machine-learning-based methods with IMU that were calibrated using gait data was high, i.e., more than 95% [19,20,21].

However, our proposed system has several advantages. The proposed system requires only one sensor, which enables the users to easily attach the sensor to their legs. Because the amount of data did not significantly affect the accuracy, the required amount of muscle deformation data for each posture was 10 s in this study. The calibration could be performed by simply standing in six postures. As the time to put on of the sensor was less than 10 s, the total preparation time for using gait phase detection was less than 70 s. Therefore, the proposed method is valid for daily use for gait phase detection.

The classification algorithm is a computational method for drawing a decision boundary in the data space. SVM applies the concept of margin maximization to determine the decision boundary. The margin is the distance between the decision boundary and the data, and margin maximization involves finding the boundary that maximizes the distance between the boundary and the data point closest to the boundary, called the support vector. The decision boundary can be obtained so that the classification performance can be improved to some extent while allowing for classification errors. Thus, the classification is robust against slight differences between standing and walking data. However, it is necessary to adjust the penalty weight to the error and the scale of the data as hyperparameters. Because the values of the hyperparameters varied among the subjects, it is difficult to achieve the same level of accuracy in this experiment in an actual-use environment where cross-validation is impossible. Logistics regression, which does not require hyperparameter adjustment, had significantly lower detection accuracy than SVM. However, the accuracy of LR-a, with the addition of probability adjustment based on gyroscope data, is equal to or better than that of SVM. The adjustment probability could reduce the error in the decision boundaries between the postures and the stance/swing phases. In this study, we applied simple information about the direction of foot movement for probability adjustment. Using the IMU-based rules in [32,33] for probability adjustment might be beneficial for improving the accuracy of the proposed algorithm in the future.

We observed individual differences in the detection accuracy. In particular, the accuracy for the female subjects tended to be lower, as shown in Figure 10. However, for the six male subjects whose muscle state was easily observed by the sensor, the gait phase could be classified into the stance and swing phases with higher accuracy (approximately 94% and 86%, respectively) only with muscle deformation data. As the stance phase and swing phase durations were approximately 0.7 and 0.4 s for male subjects, respectively, the error was approximately 0.05 s for the stance and swing phases. This accuracy is similar to that reported by some previous studies [27,29], where the calibration was performed using gait data as teaching data. Our proposed method could provide highly accurate gait phase detection if the values regarding muscle deformation are obtained clearly by the sensor.

The main feature of the system is its static standing-based calibration using a muscle deformation sensor. The calf muscle deformation information is useful for obtaining the training dataset without walking. We assumed that the foot-contact/foot-off states were detected because the calf muscle deformation changed depending on whether body weight was applied to the foot or not. Therefore, the proposed system might be used in various walking environments, such as steps and uneven terrain. We think that standing-based calibration can also be applied with force myography or ultrasound. However, we think that EMG cannot be applied for this calibration method because of the inconsistency between measurement results based on force myography and EMG [38]. The main difference between the proposed system and the system using force myography [29] is the calibration method. As force myography has poor spatial resolution [39], near-infrared light might be easier to use, which was applied in the proposed method.

One of the limitations of this study is that the current sensor usability relies on individual muscle mass. Muscle deformation information tended to be difficult to obtain in three female subjects who were considered to have low muscle mass. An increase in the number of photo reflectors might be beneficial for application in diverse subjects. In addition, the arbitrary nature of postures is another limitation. Understanding the change in muscle deformation caused by slight differences in foot position during standing in the same posture is beneficial. Furthermore, the low sample frequency of the current system limited detection accuracy. To achieve higher detection accuracy, a system should be constructed with shorter processing time and faster communication speed. In this experiment, the model did not need to include a directional extraction calculation because the anterior direction of the body (east direction) was constant and known. In practice, because people walk in any direction, this calculation will be necessary in the future.

## 5. Conclusions

In this study, we constructed a novel gait phase detection system with static standing-based calibration using muscle deformation information. A logistic regression algorithm was used as the machine learning algorithm, and the probability output was adjusted based on the angular velocity of the sensor. An experiment was performed with 10 subjects, and the detection accuracy of foot-contact and foot-off states was evaluated using video data for each subject. The median accuracy was 90% during walking based on calibration for 60 s, which demonstrated the feasibility of the static standing-based calibration method using muscle deformation information as an alternative and fast calibration method using walking data for foot-contact and foot-off state detection. There were individual differences in detection accuracy. In particular, the accuracy for the female subjects tended to be lower. Conversely, for six male subjects whose muscle state was easily observed by the sensor, the gait phase could be classified into the stance and swing phases with higher accuracy (approximately 0.05 s error) using only muscle deformation data. When the muscle deformation information was obtained clearly by the sensor, the proposed system produced similar detection accuracy as some previous studies where the calibration was performed using gait data as teaching data.

An increase in the number of photo reflectors would be beneficial for application in diverse subjects in the future. In addition, understanding the change in muscle deformation caused by slight differences in foot positioning during standing in the same posture wold be beneficial. Furthermore, the low sample frequency of the current system is limiting detection accuracy. To increase detection accuracy, a system should be constructed with shorter processing time and faster communication speed. In this experiment, the model did not need to include a directional extraction calculation because the anterior direction of the body (east direction) was constant and known. In practice, because people walk in any direction, this calculation will be necessary in the future.

## Figures and Tables

**Figure 1 sensors-21-01081-f001:**
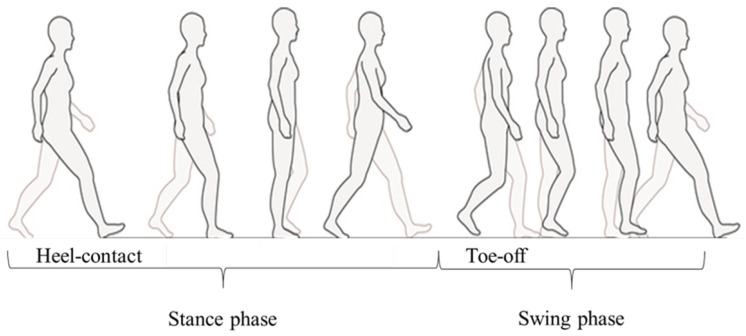
Gait phase. The stance phase starts from heel contact, and the swing phase starts from toe-off.

**Figure 2 sensors-21-01081-f002:**
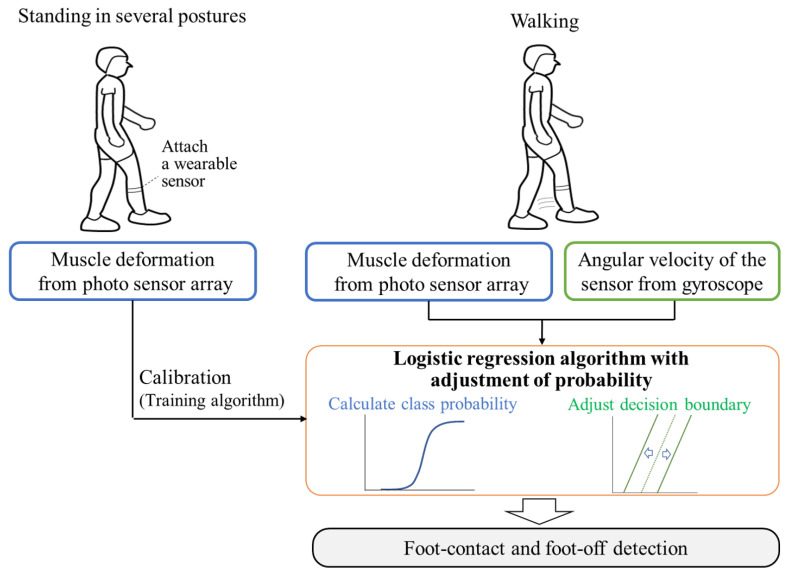
Overview of the proposed system for detecting gait phase with muscle deformation and sensor’s angular velocity based on static standing-based calibration.

**Figure 3 sensors-21-01081-f003:**
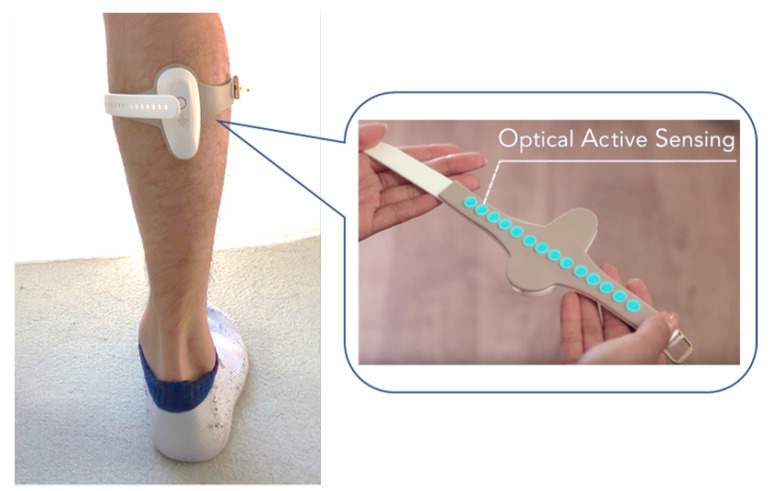
The muscle deformation sensor FirstVR attached with a shank for sensing muscle deformation information.

**Figure 4 sensors-21-01081-f004:**
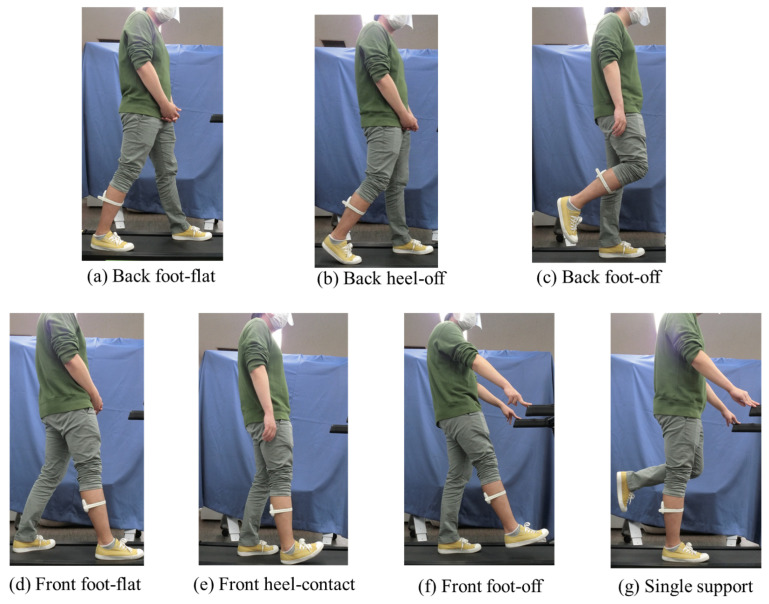
Postures for calibration of gait phase detection algorithm.

**Figure 5 sensors-21-01081-f005:**
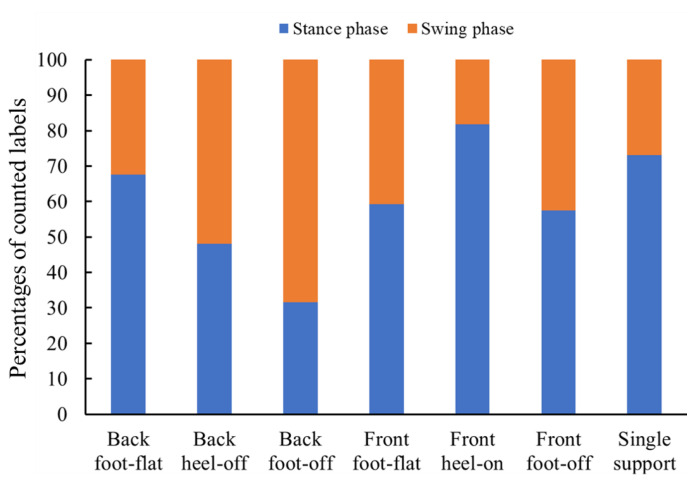
Percentages of counted labels whose muscle deformation vectors were closest to those of each posture in each gait cycle.

**Figure 6 sensors-21-01081-f006:**
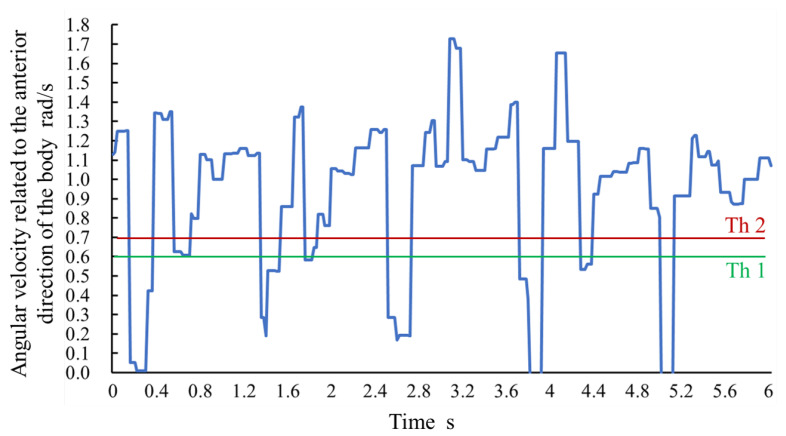
Absolute value of angular velocity related to the anterior direction of the body during walking.

**Figure 7 sensors-21-01081-f007:**
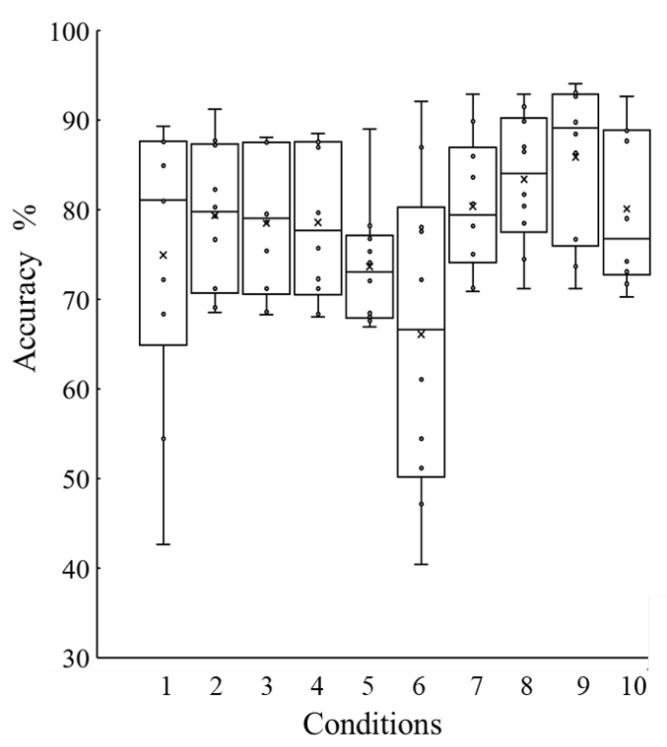
Box plot of the accuracy of foot-contact and foot-off detection for each training data combination. Circles indicate the accuracy for each participant, and x indicates the mean value.

**Figure 8 sensors-21-01081-f008:**
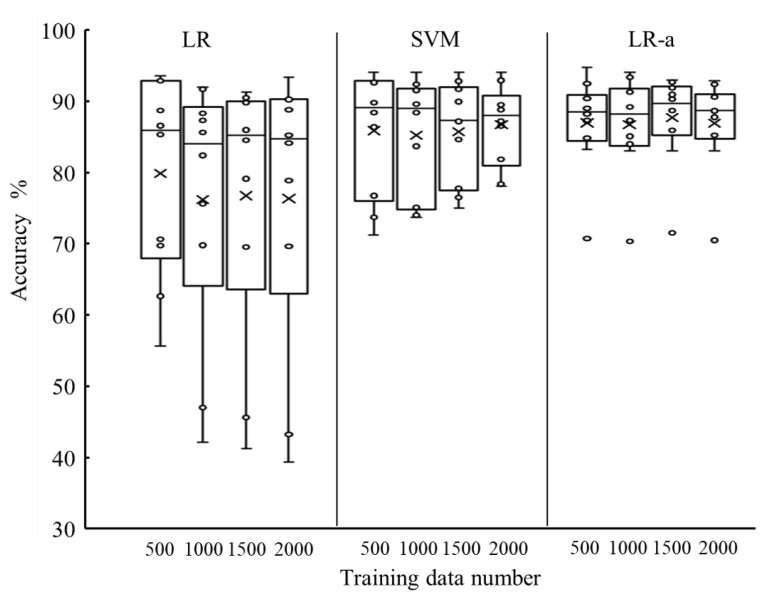
Box plot of accuracy when using logistic regression (LR) (left), support vector machine (SVM) (middle), and adjusted LR (LR-a) (right) with 500, 1000, 1500, and 2000 training data for each posture. Circle indicates accuracy for each participant, and x indicates mean value.

**Figure 9 sensors-21-01081-f009:**
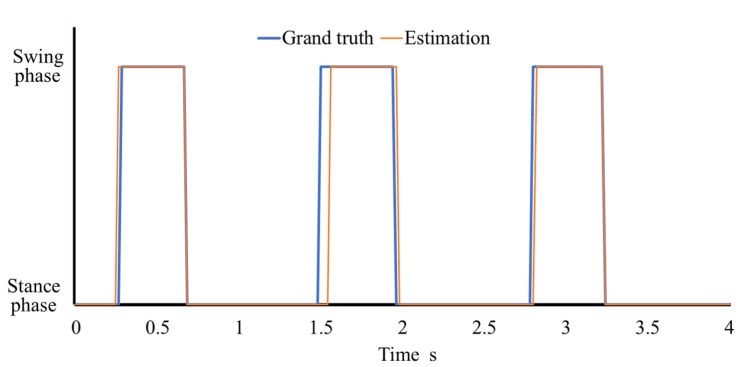
An example a time-series result of gait phase detection of a male subject using LR-a.

**Figure 10 sensors-21-01081-f010:**
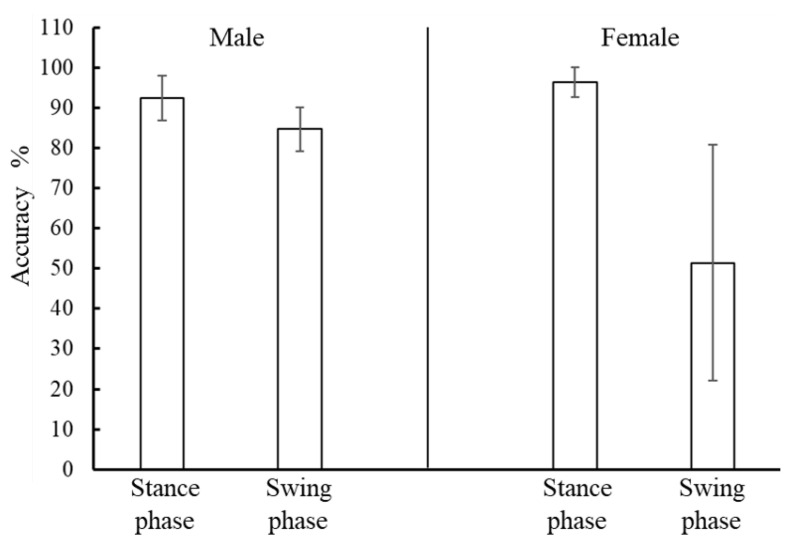
Positive rates of the stance and swing phases using LR-a with 500 training data for men (**left**) and women (**right**). The box height indicates the mean value and the bar indicates the standard deviation.

**Table 1 sensors-21-01081-t001:** The experimental conditions of combinations of standing data used as training data for calibration.

Condition	Swing Postures		Stance Postures				
	Back Foot-Off	Front Foot-Off	Front Heel-On	Single Support	Back Foot-On	Front Foot-On	Back Heel-Off
1	used	-	used	-	-	-	-
2	used	-	used	used	-	-	-
3	used	-	used	used	used	-	-
4	used	-	used	used	used	used	-
5	used	-	used	used	used	used	used
6	used	used	used	-	-	-	-
7	used	used	used	used	-	-	-
8	used	used	used	used	used	-	-
9	used	used	used	used	used	used	-
10	used	used	used	used	used	used	used

**Table 2 sensors-21-01081-t002:** True positive rates of the stance and swing phases detected by LR-a. IQR, interquartile range.

Training Data Number	True Positive Rate of Stance Phase %		True Positive Rate of Swing Phase %	
	Median	IQR	Median	IQR
500	95	6	81	25
1000	93	10	84	17
1500	95	6	79	17
2000	93	11	81	21

## Data Availability

Not applicable.

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
