# Peer review of "Gait Phase Detection Based on Muscle Deformation with Static Standing-Based Calibration"

_sensors, 2021, doi:10.3390/s21041081_

Round 1

Reviewer 1 Report

In this paper, the authors present a novel system for gait phase detection. The proposed system uses muscle deformation information to obtain the accuracies of foot-contact and foot-off states. The authors carried out experiments with 19,651 frames obtained from 10 subjects.

In general, the scientific quality of the paper and its relevance in the field is good. This paper is very interesting. The proposed methodology is clear and described in adequate detail. The figures are adequate and the experimental results show that the proposed method has a similar precision to the methods in the state of the art.

reference 36 is incomplete and/or incorrect.

Author Response

General comment: In this paper, the authors present a novel system for gait phase detection. The proposed system uses muscle deformation information to obtain the accuracies of foot-contact and foot-off states. The authors carried out experiments with 19,651 frames obtained from 10 subjects. In general, the scientific quality of the paper and its relevance in the field is good. This paper is very interesting. The proposed methodology is clear and described in adequate detail. The figures are adequate and the experimental results show that the proposed method has a similar precision to the methods in the state of the art.

Answer:

 We would like to thank this reviewer for having carefully read our study. In the following sections, we replied to all concerns. Also, we highlighted how the Manuscript was changed. Also, we uploaded the answer sheet as a document file. Please see the attachment.

Q1.  reference 36 is incomplete and/or incorrect.

A1. We added the more information to the Reference 37.

Reviewer 2 Report

It is an interesting idea to use static standing-based calibration to avoid labeling work.

There are more work on IMU for gait analysis (non machine learning work) can be cited and may be interesting to this investiage.  For example, the gait dynamic image feature of gyroscope provide a way to detedtion walking pace ( "Sensor orientation invariant mobile gait biometrics." IJCB 2014, and "Pace independent mobile gait biometrics" BTAS 2015).  The GDI feature may provide a stronger feature than angular velocity alone for foot-contact and foot-off classification.

Could some feature selection or search method be applied to get a more exhustive search for all possible expeerimental condisionts listed in Table 1?

The test data were collected on treadmill. Do you exptect the resluts to be hold when used off the readmill, where the walking pace is not constrained to be a constant?

Author Response

General comment: It is an interesting idea to use static standing-based calibration to avoid labeling work.

Answer:

We would like to thank this reviewer for having carefully read our study and providing us with useful suggestions. In the following sections, we replied to all concerns. We also uploaded a document file. Please see the attachment.

Q1. There are more work on IMU for gait analysis (non machinelearning work) can be cited and may be interesting to this investiage.  For example, the gait dynamic image feature ofgyroscope provide a way to detedtion walking pace ( "Sensororientation invariant mobile gait biometrics." IJCB 2014, and "Paceindependent mobile gait biometrics" BTAS 2015).  The GDI featuremay provide a stronger feature than angular velocity alone for foot-contact and foot-off classification.

A1. We added the paragraph explaining previous research of non-machine-learning methods including GDI in the 3rd paragraph of the Section 1.2. Previous research.

There are several approaches for gait phase detection without obtaining training data. An auto-calibration method using a planar structure in the angular space of leg joints has been proposed in [31] to detect heel-contact and toe-off. However, this method requires expert knowledge to attach expensive three joint angle sensors to the subject; thus, increasing the cost, time of attachment, and complexity of use. It is better to develop a high-usability gait phase detection system with few sensors. An IMU-based system could apply rule-based detection algorithm based on thresholds or searching of minimum and maximum values, which do not require training data [32, 33]. For instance, a zero-crossing detection approach finding positive and negative peaks of angular velocity detected gait events within error of 103 ms [32]. An IMU-based system with a rule-based algorithm using dynamic image features was used for identification of gait biometrics [34]. Rule-based methods were limited to inertial information among wearable devices, and other sensing information might require machine learning because the sensor readings vary due to differences in the attachment condition of the sensor. Using different sensing information for gait phase detection without manual labeling of the gait data is beneficial to increase the application scenarios or to contribute to future fusion of sensing information for better detection accuracy.

Q2. Could some feature selection or search method be applied to get a more exhustive search for all possible experimental condisionts listed in Table 1?

A2. We assume that feature extraction or search method is generally applied to determine which of the data obtained from each channel of FirstVR can be used to improve accuracy rather than a combination of the training dataset. Maybe these methods could also be applied for determining combinations of training dataset, but we assumed that we needed not to use such methods to find a possible combination of training datasets as experimental conditions. This is because the relevance degree between muscle deformation in each gait phase and in each standing training dataset (standing posture) is expressed by the Euclidean distance. We investigated this parameter in Figure 5.

Q3. The test data were collected on treadmill. Do you exptect the resluts to be hold when used off the treadmill, where the walking pace is not constrained to be a constant?

A3. Yes, we assume that the proposed method can be applied for walking off the treadmill. In this experiment, the subjects walked at their preferred speed (2.93±0.26 km/h). Therefore, the proposed method could work at least within the margin of 2.93±0.26 km/h. We assume that the muscle deformation based partition identify whether user’s weight is applied to the foot or not, and over-ground walking and treadmill walking are not different so much in terms of weight application to the foot. However, further reasearch how much difference in speed the proposed algorithm can handle will be required.

Reviewer 3 Report

This paper presents an interesting new technique for gait cycle identification based on using optical sensors for measuring muscle deformation. In general, the procedure is clearly described, however, the claims regarding existing systems are not well justified. I would recommend that the focus be changed to the description of a novel technique, as well as expanding on where this technique can be more useful than other sensing options, rather than making unsupported claims that this technique is better than techniques using other sensors.

There are many grammatical errors throughout the manuscript, it requires additional English editing

Abstract: lines 4-6: this seems to be a "straw-man" argument (i.e. not well supported) - there are many studies showing success in identifying gait events without "precise training", and there is no significant theoretical advance made here that justifies this difference

Line 27: "floor force" - usually described as ground reaction force, and if you are talking about the gold standard, it would be ground reaction force as measured using a force plate.

Line 37: "threshold-based usually have no adaptability" - this is not the case - the threshold can just be changed- as is described in the following lines

Line 53: "most of these techniques" - some papers use machine learning, while others define rules, e.g. see

J. C. Pérez-Ibarra, A. A. G. Siqueira and H. I. Krebs, "Real-Time Identification of Gait Events in Impaired Subjects Using a Single-IMU Foot-Mounted Device," in IEEE Sensors Journal, vol. 20, no. 5, pp. 2616-2624, 1 March1, 2020, doi: 10.1109/JSEN.2019.2951923.

As they use "rules" rather than ML, they do not require extensive training data

Line 96: Please describe the particular sensor used (e.g. model number, year of manufacture)

Line 149: Usually the magnetometer is insufficient to provide Euler angles, rather an IMU is required. Are you sure that only a magnetometer is used here?

Line 150: The equation is missing closing brackets

Line 171: Please be more specific about where the sensor was placed - the shank is relatively long (compared to the sensor)

Line 202-203: Please expand on how the gait events were identified from the video

Line 203-204: which sensors were used / not used? Why?

Line 270-271: At this point, the results should be compared to other studies in the literature.

Line 294 - if sex differences are to be discussed, the results should be presented (I could not find them)

Line 307 - It would be good to compare this system to systems using "force myography" - what are the advantages / disadvantages of the two techniques (which are somewhat related), these systems have also been used for gait detection

Figures 7&8 - you can remove the white space (low %), and it would be helpful to put the graphs side by side to allow visual comparison between them. Also, given the small number of subjects, why not just present all data (as well as the mean), maybe with male and female in different colors

Figure 8: How are the 3 algorithms (LR, SVM and LR-A) shown in this graph? It looks like results from one method

References: be consistent in journal abbreviations (e.g. either Gait Posture, or Gait & Posture)

Reference 36: reference to a website is missing the URL

Reference 37: incomplete (is this a book? Article? Website?)

Line 356: Conflict of interest - the fact that one of the authors works for H2L who built the hardware should be listed here

Line 450: Sample availability - this should be made more specific - what is made available? Ideally the authors would upload the analysis software to a website such as figshare

Author Response

General comment: This paper presents an interesting new technique for gait cycle identification based on using optical sensors for measuring muscle deformation. In general, the procedure is clearly described, however, the claims regarding existing systems are not well justified. I would recommend that the focus be changed to the description of a novel technique, as well as expanding on where this technique can be more useful than other sensing options, rather than making unsupported claims that this technique is better than techniques using other sensors.

Answer:

 We would like to thank this reviewer for having carefully read our study and providing us with useful suggestions. Mainly, we modified the focus, as pointed out. In the following sections, we replied to all concerns and highlighted how the Manuscript was changed. We also uploaded the document version of the answerers. Please see the attachment.

Q1. There are many grammatical errors throughout the manuscript, it requires additional English editing

A1. English of the manuscript was revised again.

Q2. Abstract: lines 4-6: this seems to be a "straw-man" argument (i.e. not well supported) - there are many studies showing success in identifying gait events without "precise training", and there is no significant theoretical advance made here that justifies this difference

A2. We removed this argument in Abstract and the whole manuscript.

Q3. Line 27: "floor force" - usually described as ground reaction force, and if you are talking about the gold standard, it would be ground reaction force as measured using a force plate.

A3. We used a camera system which is one of gold standards. Therefore, we modified this sentence to mention only the camera system.

Q4. Line 37: "threshold-based usually have no adaptability" - this is not the case - the threshold can just be changed- as is described in the following lines

A4. As pointed out, this sentence is not appropriate, thus we removed it.

Q5. Line 53: "most of these techniques" - some papers use machine learning, while others define rules, e.g. see J. C. Pérez-Ibarra, A. A. G. Siqueira and H. I. Krebs, "Real-Time Identification of Gait Events in Impaired Subjects Using a Single-IMU Foot-Mounted Device," in IEEE Sensors Journal, vol. 20, no. 5, pp. 2616-2624, 1 March1, 2020, doi: 10.1109/JSEN.2019.2951923. As they use "rules" rather than ML, they do not require extensive training data

A5. Because the techniques mentioned in this paragraph applied machine learning, we kept this sentence as it was, and added the rule-based methods in the 3rd paragraph of the Section 1.2. Previous research.

There are several approaches for gait phase detection without obtaining training data. An auto-calibration method using a planar structure in the angular space of leg joints has been proposed in [31] to detect heel-contact and toe-off. However, this method requires expert knowledge to attach expensive three joint angle sensors to the subject; thus, increasing the cost, time of attachment, and complexity of use. It is better to develop a high-usability gait phase detection system with few sensors. An IMU-based system could apply rule-based detection algorithm based on thresholds or searching of minimum and maximum values, which do not require training data [32, 33]. For instance, a zero-crossing detection approach finding positive and negative peaks of angular velocity detected gait events within error of 103 ms [32]. An IMU-based system with a rule-based algorithm using dynamic image features was used for identification of gait biometrics [34]. Rule-based methods were limited to inertial information among wearable devices, and other sensing information might require machine learning because the sensor readings vary due to differences in the attachment condition of the sensor. Using different sensing information for gait phase detection without manual labeling of the gait data is beneficial to increase the application scenarios or to contribute to future fusion of sensing information for better detection accuracy.

Q6.Line 96: Please describe the particular sensor used (e.g. model number, year of manufacture)

A7 We added the model number and year of manufacture in Line 102 (modelnumber:FVR-SET01,yearofmanufacture:2018).

Q7. Line 149: Usually the magnetometer is insufficient to provide Euler angles, rather an IMU is required. Are you sure that only a magnetometer is used here?

A7. In this study, only magnetometer values were used for calculation (2). We also assumed that the deviation errors of euler angles were included here. However, because only simple information about leg movement were used for probability adjustment, the required accuracy of the euler angle was low. Furthermore, we set a simple rule; the angular velocity values were unreliable when tan(θ) was significantly high  because the large error of transformation (especially tangent calculation) causes a fault in the probability adjustment. (explained in Line 156)

As a future work, we would try a more accurate algorithm deviating euler angles and rules proposed in the previous research [32, 33].

Q8. Line 150: The equation is missing closing brackets

A8. We added the closing brackets in equation 2.

Q9. Line 171: Please be more specific about where the sensor was placed - the shank is relatively long (compared to the sensor)

A9. The sensor was attached to the center of a right lower leg in the long axis direction. Therefore, we added the information about the center in this long axis direction in Line 179-180.

Q10. Line 202-203: Please expand on how the gait events were identified from the video

A10. We added more detail explanation how the gait events were identified from the video in Line 210-213.

Frames in which the heel of the right foot contacted the treadmill and the toe of the right foot left the treadmill were extracted by visual observation. The frames from the toe-off to the heel-contact and from the heel-contact to the toe-off were labeled as the swing phase and stance phase, respectively.

Q11. Line 203-204: which sensors were used / not used? Why?

A11. I apologize that the phrase was wrong. It is explaining about channels not sensors. Therefore, we modified the sentence to “Among the 14channels of FirstVR, 11 were reacted and used as model inputs”. in Line 214-215)

The other 3 channels did not work.

Q12. Line 270-271: At this point, the results should be compared to other studies in the literature.

A12. We added sentences for comparison with other studies in the literature in Line 283-288.

Although one of the previous rule-based methods with IMU [33] did not evaluate sample based accuracy, the detection accuracy of such methods might be same as this work considering absolute mean error and ratio of strides where gait events were detected to all strides in the researches [32, 33]. The literature using machine learning-based methods reports higher accuracy than the rule-based methods and this work. Especially, the accuracy of machine learning-based methods with IMU which were calibrated using gait data was high; i.e. more than 95 % [19-21].

Q13. Line 294 - if sex differences are to be discussed, the results should be presented (I could not find them)

A13. We added a new figure showing the difference of accuracy for male and female as Figure 10 to show clearly different trends of detection accuracy between male and female.

Q14. Line 307 - It would be good to compare this system to systems using "force myography" - what are the advantages / disadvantages of the two techniques (which are somewhat related), these systems have also been used for gait detection

A14. We added sentences mentioning comparison with the system using force myography Line 310-313.

In this study, we applied the simple information about direction of foot movement for probability adjustment. Using the IMU-based rules of [32,33] for probability adjustment might be beneficial for improving the accuracy of the proposed algorithm in the future.

Q15. Figures 7&8 - you can remove the white space (low %), and it would be helpful to put the graphs side by side to allow visual comparison between them. Also, given the small number of subjects, why not just present all data (as well as the mean), maybe with male and female in different colors

A15. We removed the lower white space of figures 7&8. We considered how to modify figures and assume that it would be better to keep separating figures 7&8. In analysis of this work, firstly the training data combination required for gait phase detection was investigated, and then the effect of algorithms and training data number were investigated using the best condition of the training data combination. Therefore, Figure 8 is based on the result shown in Figure 7.

Box plot includes all participant data. Original version shows the only minimum, first quartile, median, mean, second quartile, and maximum values. We added circle plots to let readers read the data of all participants. To make the difference of detection accuracy between male and female clearer, we added a new figure (Figure10).

Q16. Figure 8: How are the 3 algorithms (LR, SVM and LR-A) shown in this graph? It looks like results from one method

A16. 4 conditions of training data number (500, 1000, 1500, 2000) for each algorithm (LR, SVM and LR-A from left to right) were shown in this graph. Because it was difficult to see as pointed out, we modified the graph for better readability. Algorithms were partitioned with vertical line and algorithm name was written in the upper side in the new figure 8.

Q17. References: be consistent in journal abbreviations (e.g. either Gait Posture, or Gait & Posture)

A17. We unified the journal abbreviations as Gait & Posture.

Q18. Reference 36: reference to a website is missing the URL.

A18. We added the URL in Reference 36.

Q19. Reference 37: incomplete (is this a book? Article? Website?)

A19. We added more information in Reference 37.

Q20. Line 356: Conflict of interest - the fact that one of the authors works for H2L who built the hardware should be listed here

A20. We added a sentence below;

Hosono, one of the authors, belongs to the R\&D department of H2L, the company that developed FirstVR. He contributed to the programming part of our research.

Q21. Line 450: Sample availability - this should be made more specific - what is made available? Ideally the authors would upload the analysis software to a website such as figshare

 A21. Because we have not decided the detail method, we just explained  “The data used to support the findings of this study are available from the corresponding author upon request.”

Round 2

Reviewer 3 Report

I am satisfied with the corrections made by the authors